# Treatment of Swine Closed Head Injury with Perfluorocarbon NVX-428

**DOI:** 10.3390/medsci8040041

**Published:** 2020-09-27

**Authors:** Francoise Arnaud, Ashraful Haque, MAJ Erin Morris, Paula Moon-Massat, Charles Auker, Saha Biswajit, Brittany Hazzard, Lam Thuy Vi Tran Ho, Richard McCarron, Anke Scultetus

**Affiliations:** 1Naval Medical Research Center, NeuroTrauma Department, 503 Robert Grant Ave, Silver Spring, MD 20910, USA; ashraful.haque.mil@mail.mil (A.H.); pfmoonmassat@outlook.com (P.M.-M.); chipra@aol.com (C.A.); biswajit.saha.ctr@usuhs.edu (S.B.); bhazzard100@gmail.com (B.H.); vi.tranho@gmail.com (L.T.V.T.H.); dickmccarron@gmail.com (R.M.); anke.h.scultetus2.civ@mail.mil (A.S.); 2The Henry M. Jackson Foundation for the Advancement of Military Medicine Inc., 6720 Rockledge Drive, Bethesda, MD 20817, USA; 3Department of Surgery, Uniformed Services University of the Health Sciences, 4301 Jones Bridge Rd., Bethesda, MD 20814, USA; 4Walter Reed Army Institute of Research, Veterinary Pathology Services, 503 Robert Grant Ave, Silver Spring, MD 20910, USA; erin.k.morris4.mil@mail.mil

**Keywords:** neuroprotection, oxygen therapeutic, dodecafluoropentane, perfluorocarbon, brain trauma

## Abstract

Pre-hospital treatment of traumatic brain injury (TBI) with co-existing polytrauma is complicated by requirements for intravenous fluid volume vs. hypotensive resuscitation. A low volume, small particle-size-oxygen-carrier perfluorocarbon emulsion NVX-428 (dodecafluoropentane emulsion; 2% *w*/*v)* could improve brain tissue with minimal additional fluid volume. This study examined whether the oxygen-carrier NVX-428 shows safety and efficacy for pre-hospital treatment of TBI. Anesthetized swine underwent fluid percussion injury TBI and received 1 mL/kg IV NVX-428 (TBI-NVX) at 15 min (T15) or normal saline (no-treatment) (TBI-NON). Similarly, uninjured swine received NVX-428 (SHAM-NVX) or normal saline (SHAM-NON). Animals were monitored and measurements were taken for physiological and neurological parameters before euthanasia at the six-hour mark (T360). Histopathological analysis was performed on paraffin embedded tissues. Physiological, biochemical and blood gas parameters were not different, with the exception of a significant but transient increase in mean pulmonary artery pressure observed in the TBI-experimental group immediately after drug administration. There were no initial differences in brain oxygenation at baseline, but over time oxygen decreased ~50% in both TBI groups. Histological brain injury scores were similar between TBI-NVX and TBI-NON, although a number of subcategories (spongiosis-ischemic/dead neurons-hemorrhage-edema) in TBI-NVX had a tendency for lower scores. The cerebellum showed significantly lower spongiosis and ischemic/dead neuron injury scores and a lower number of Fluoro-Jade-B-positive cerebellar-Purkinje-cells after NVX-428 treatment compared to controls. NVX-428 may assist in mitigating secondary cellular brain damage.

## 1. Introduction

Traumatic brain injury (TBI) produces alterations in cerebrovascular function, edema, ischemia, and brain tissue hypoxia [1,2,3]. Early TBI treatment may improve outcome [4] as well as benefit patients suffering from other neurological disorders (e.g., stroke). However, treatment of TBI with concurrent hypovolemia in polytrauma patients is complicated by potentially conflicting requirements for fluid resuscitation (normo- vs. hypotensive resuscitation). A low volume oxygen carrier could address this dilemma by improving brain oxygen delivery. The oxygen-carrying perfluorocarbon NVX-428 (NuvOx Pharma, Tucson, AZ, USA) may be a promising compound for pre-hospital TBI treatment as clinical and pre-clinical results indicate that NVX-428 improves tissue oxygen during hypoxic events [5,6,7,8,9,10,11].

The active component of NVX-428 is dodecafluoropentane (DDFP; 2% *w*/*v*). DDFP is stabilized into an emulsion (DDFPe = NVX-428) with surfactant in a buffered sucrose solution. DDFP has a short circulation half-life in vivo in human and rabbits (1–3 min) and is eliminated through exhalation from the lungs (half-life of 90 min) consistent with sustained (>90 min) therapeutic oxygen transport effects [6,8]. Uniquely, when NVX-428 is at room temperature, it consists of stabilized nanodroplets suspended in emulsion (mean particle size of ~250 nm) and DDFP remains in this condensed state during intravenous (IV) administration [8,12,13]. At body temperature, the droplets are postulated to expand slightly, providing enhanced oxygen transport ability through the formation of CF_3_ group cavities within the liquid that act as a “respiratory sink” for surrounding oxygen to dissolve into [13,14]. The doses of NVX-428 for use as an oxygen therapeutic are less than one one-hundredth of other perfluorocarbons on a weight basis [15]. Larger bubbles can manifest as gas embolism (decompression sickness), but NVX-428 droplets are small enough to pass through capillary beds without adverse effects [16]. We previously evaluated NVX-428 for TBI in healthy rats, and found no effect on systemic blood pressure nor change in cerebral pial arteriole diameters [17]. Brain tissue oxygenation (PbtO_2_) also improved when NVX-428 was administered after TBI in rats [10].

NVX-428 was previously tested in humans at low doses (0.05 mL/kg) as an ultrasound contrast agent and was shown to be safe [16]. The drug product has also been shown to be safe at a dose of 0.17 mL/kg administered three times 90 min apart in a Phase Ib/II trial (NCT02963376) in acute ischemic stroke patients [18] and at a dose of 0.10 mL/kg administered once daily, five days per week for six weeks prior to each fraction of radiation in a Phase Ib/II trial (NCT02189109) in glioblastoma multiform patients [19]. These data and product characteristics suggest that NVX-428 may be suitable for the pre-hospital environment, particularly in areas where refrigeration is not immediately available, such as in military conflicts.

This study sought to determine if safety and efficacy, similar to our rat study, could be reproduced in a swine model of fluid-percussion injury (FP)-TBI. The hypotheses were that NVX-428 would reduce histological evidence of brain injury and improve physiological parameters (e.g., intracranial pressure [ICP]) when compared to no therapy.

## 2. Results

All 22 swine survived to the end of the experiment at T360. There were no differences in baseline hemodynamic parameters or fluid percussive force producing the brain injury for TBI-NVX and TBI-NON groups (45.9 ± 1.5 psi and 45.9 ± 1.9 psi, respectively). Nine of the 14 TBI pigs required mannitol for ICP > 20 mm Hg with no differences between NVX and NON groups.

### 2.1. Hemodynamics

Most hemodynamic changes were the result of the TBI and not the drug treatment (Figure 1). There were differences in mean arterial pressure (MAP), heart rate (HR), cardiac index (CI), central venous pressure (CVP), pulmonary and systemic vascular resistance indexes (PVRI) and systemic vascular resistance index (SVRI) between TBI and SHAM-groups after injury but none between NVX and NON. From T15 to T75, HR and CI were higher and MAP, CVP and SVRI lower immediately after TBI vs. SHAM uninjured swine due to the brain injury (*p* < 0.01). These post-TBI HR changes lasted for approximately two hours before returning toward baseline levels. After infusion of NVX-428 in the TBI group, HR increased less compared to NON with saline treatment (*p* < 0.05).

The only clinically relevant treatment-related hemodynamic effect was a rapid, transient increase in mean pulmonary artery pressure (MPAP) following the injection of NVX-428 (Figure 2). Compared to baseline, MPAP changed within the TBI-NVX group after NVX treatment. Specifically, compared to T0, MPAP was lower at T15 and higher at T30 and T240. Although a similar trend was visually apparent in the SHAM-NVX group, the increase in MPAP was not as large and was not significant (*p* = 0.056; Figure 2). There were no differences in overall MPAP between TBI-NVX vs. TBI-NON or between SHAM-NVX vs. SHAM-NON, following the particularly transient nature of the MPAP increases.

### 2.2. Oxygen and Metabolism

PbtO_2_ was similar in all groups at T0 30.8 ± 2.7 mmHg but decreased from 30.3 ± 3.9 mmHg at T0 to 18.1 ± 3.1 mmHg at T15 in the TBI groups and tended to remain lower than in the SHAM groups (Figure 3). There was no difference for PbtO_2_ between NVX and NON groups for the remaining course of the experiment. At T15, cerebral perfusion pressure (CPP) decreased and intracranial pressure (ICP) increased from baseline (Figure 4) in the TBI groups. After TBI, CPP decreased and stayed lower throughout the study. ICP was not different between injury groups (TBI-NVX vs. TBI-NON) or between the uninjured groups (SHAM-NVX vs. SHAM-NON). ICP increased in both TBI groups and remained elevated while ICP rose in both SHAM groups near the end of the study (Figure 4).

Neither TBI nor NVX-428 had any effect on hematologic or blood oxygen related gas parameters (PO_2_, PCO_2_) regardless of the group. There was also no difference in oxygen delivery (DO_2_) or consumption (VO_2_) across the groups. The only exception was significantly higher hemoglobin (Hb) and lactate (lac) in the TBI groups compared to SHAMs at T15 (Figure 4). This might be an effect of the TBI as this also returned to baseline within 2 h.

### 2.3. Histopathology

As expected, histology total brain injury scores were higher in the group with brain injury (TBI-NON) compared to the uninjured group (SHAM-NON) (Figure 5a), with the TBI-NON having higher scores for spongiosis and ischemic neurons than SHAM-NON (Figure 5b). In addition, when the cerebral cortex, cerebellum and hippocampus were analyzed for spongiosis and for presence of ischemic/dead neurons (Figure 5c,d), the TBI-NON groups had more spongiosis in the cerebellum and hippocampus and more ischemic neurons in the cerebellum and cerebral cortex than the SHAM-NON groups(*p* < 0.01). Thus, the fluid percussion TBI model produced measurable evidence of injury consistent with ischemic/hypoxic damage. Presence of cerebral spongiosis in the SHAM-NON indicates that surgical placement of the fluid percussion device introduces some mild damage. The addition of NVX-428 showed a significant t reduction of the cerebellum spongiosis and dead neurons after TBI (*p* < 0.05).

To evaluate the effects of NVX-428 on head injury, the most important comparisons were between brain-injured animals that were treated (TBI-NVX) vs. untreated (TBI-NON). A total brain injury score (Figure 5a) was calculated by adding each pathology score from each subregion of the brain for each group. This injury score was not different between the NON and NVX groups. Certain pathologies were more developed such as spongiosis and presence of ischemic neurons in TBI vs. SHAM groups (*p* < 0.05) with a trend for lower score when NVX-428 was used (Figure 5b). There were differences for certain brain regions (cerebrum, hippocampus and cerebellum) (Figure 5c,d) known to be selectively vulnerable to hypoxia [2]. When these regions were evaluated more specifically, the cerebellum, in particular, showed less injury after TBI when treated with NVX-428 for spongiosis and ischemic/dead neuronal pathology (Figure 5, Figure 6 and Figure 7). There were no differences in Fluoro-Jade B positive cells between the TBI-NVX and TBI-NON groups, but the mean number of Fluoro-Jade B positive cerebellar Purkinje cells was lower in the TBI-NVX group compared to the TBI-NON group (1.3 ± 0.7 vs. 5.3 ± 1.8 cells/field, respectively; *p* < 0.05; Figure 6). The hippocampus and cerebrum did not show a difference between TBI-NVX and TBI-NON in the amount of Fluoro-Jade B positive cells or their severity scores.

## 3. Discussion

In this study, NVX-428 treatment of TBI resulted in reduced histopathological evidence of brain injury when compared to no therapy in an acute, moderate-severity TBI swine model. Brain tissue hypoxia following TBI is at least in part due to ischemia, although diffusion barriers may also contribute [20]. Factors that may add to cerebral ischemia from TBI include: (1) decreased perfusion pressure, due to decreased arterial pressure, edema, or both, or (2) trauma-induced cerebrovascular dysfunction [1,21,22]. In our model, TBI increased ICP and decreased CPP; we did not assess cerebrovascular reactivity.

Among our results, spongiosis, an etiologically non-specific finding consistent with an early response to hypoxia, was found in the hippocampus and more prominently in the cerebellum. More specific to a hypoxic etiology, red, shrunken, pyknotic Hematoxylin and Eosin (H&E) stained neurons were found in the cerebral cortex, and more plentifully in cerebellar Purkinje neurons of brain-injured animals. A non-significant trend toward a similar finding was also discernible in the hippocampus neurons. Furthermore, Fluoro-jade B staining of cerebellar Purkinje cells was observed following TBI, a stronger indicator of irreversible cell death than H&E findings [23,24]. Cerebellar Purkinje neurons are among the neuronal types that are known to be selectively vulnerable to hypoxia. It is unclear why other selectively vulnerable neuronal populations, in particular hippocampal CA1 neurons and cerebrocortical pyramidal neurons, were not as clearly affected in our model. Possible explanations include regionally differential effects on blood flow following TBI as administered to our animals, neuronal type differential time-courses for post-injury development of histopathological changes [25], or species-specific differences in neuronal selective vulnerability.

Despite little to no differences in the physiological parameters (transient MPAP increase) after NVX-428 treatment, including no improvement in ICP or CPP, there was clear histological evidence of a neuroprotective effect with NVX-428 treatment. Noteworthy, the transient MPAP could be a result of pulmonary intravascular macrophages (PIMs) in swine, which instantaneously react to injected nanoparticles leading to periods of peak vasoconstriction, bronchoconstriction and pulmonary hypertension [26]. However, a direct comparison in humans would be necessary to discern this hypothesis from a true product effect. The degree and severity of brain injury in regions of the brain most sensitive to ischemia were reduced. In particular, NVX-428 mitigated TBI-induced cerebellar spongiosis and ischemic/dead neurons and decreased the number of Fluoro-Jade B positively labeled Purkinje cells. There was also a non-significant trend for SHAM-NVX brain injury scores to be lower than the other three groups. Brain surgical instrumentation, itself necessary to place the fluid percussion device, produced a low degree of brain injury as seen in the brain pathologies (Figure 5b) where the scores in NON groups are above baseline. The tendency for SHAM-NVX to have the lowest injury scores is supportive of the beneficial effects of NVX-428 even with mild trauma. The results of this traumatic brain injury study indicate that NVX-428 may be a medically useful oxygen carrying compound under a variety of non-traumatic conditions including stroke, myocardial infarction, anemia, and hypoxic solid tumors [5,6,7,9,11,19,27]. One limitation of this study is the absence of a naïve control group. However, it is reasonable to assume that NVX-428 is safe as the animals in the SHAM group exhibited better pathology scores.

One interesting aspect of NVX treatment is the drug’s short half-life and rapid excretion via the lungs [8]. The positive effect of the drug, however, has a longer duration of action as its effects appeared to last for 90 min in a model of stroke [8] and, in our own cortical impact TBI rat study, the improvement in brain oxygenation also lasted for at least 90 min [10]. The transient increase in MPAP seen in the current study may be a direct effect of the drug in the pulmonary circulation while it is more prolonged, clinically beneficial effects may be due to deposition of the drug into tissues. In addition, unlike humans, pigs have pulmonary intravascular macrophages (PIMs), which instantaneously react to injected nanoparticles leading to periods of peak vasoconstriction, bronchoconstriction and pulmonary hypertension [26]. Alternatively, an active metabolite may be produced although this has not been confirmed. Thus, additional research is required to clarify the mechanism of this drug’s actions in ischemic conditions and the disparity between circulating half-life and clinical duration of action.

This study was limited by the lack of more specific PbtO_2_ data which would have enabled identification of a possible mechanism for the neuroprotective effect. Nonetheless, we previously observed increased brain oxygenation following NVX-428 treatment in brain-injured rats using a phosphorescent quenching method to measure PbtO_2_ [10]. While it is likely that this occurred in the current swine study, there were several differences between studies that prevent this extrapolation from being definitive. There are differences in the type and severity of TBI (CCI vs. FP) and the response of animal species (rats vs. swine). Confirmation from future swine studies involving NVX-428 while using a different PO_2_ measuring technique is necessary.

## 4. Material and Methods

The study protocol was reviewed and approved by the Walter Reed Army Institute of Research/Naval Medical Research Center Institutional Animal Care and Use Committee in compliance with all applicable Federal regulations governing the protection of animals in research. The experiments reported herein were conducted in compliance with the Animal Welfare Act and per the principles set forth in the “Guide for Care and Use of Laboratory Animals”, Institute of Laboratory Animals Resources, National Research Council, National Academy Press, 2011.

### 4.1. Animal Preparation

Yorkshire swine (Animal Biotech Industries, Danboro, PA, USA; 31.5 ± 0.7 kg [mean ± SEM]; N = 22) were fasted ~12 h prior to the experiment with water provided ad libitum. Anesthesia was induced with ketamine/atropine and maintained with isoflurane in 100% oxygen for surgical instrumentation after which it was switched to propofol, midazolam and fentanyl for total intravenous anesthesia. Animals were mechanically ventilated in dorsal recumbence with (FiO_2_ = 0.4) (Apollo^®^ ventilator, Draeger Medical Inc. Telford, PA, USA) using synchronized intermittent mechanical ventilation (SIMV) and a positive end-expiratory pressure (PEEP) of 5 cm H_2_O. Tidal volume and respiration rate were adjusted to maintain normocapnia (35–40 mmHg) as measured by end-tidal carbon dioxide (ETCO_2_) capnography or PaCO_2_ values [28]. A femoral arterial catheter was used for blood pressure monitoring and arterial blood sampling and a femoral venous catheter was used for infusion of intravenous anesthetics and maintenance fluids. A Swan-Ganz continuous cardiac output catheter (Edwards Lifesciences LLC, Irvine, CA, USA) was used for measuring cardiac output, body temperature and for collection of mixed venous blood samples. All swine received 3 mL/kg/h normal saline as a maintenance fluid to balance fluid loss from instrumentation and the effects of general anesthesia. A Licox^®^ PMO probe was inserted in the parietal side of the brain to measure PbtO_2_ (Licox^®^ CMP Monitor; Cat #AC31; Integra LifeSciences Corp., Plainsboro, NJ, USA).

Per the manufacturing company (NuvOx Pharma, Tucson, AZ, USA), “NVX-428” designates the test drug (sonicated, 2% *w*/*v* DDFPe) for a TBI indication. The formula is identical to that studied by our laboratory under the designation “NVX-108” [10,16] except the current formula was sonicated. NVX-428 was withdrawn from its vial using a positive pressure technique, capped immediately and placed vertically before administered as IV bolus over 2 to 3 min.

### 4.2. Injury

The animals in the TBI group were subjected to an insult via fluid percussion instrument (FPI) [28]. Briefly, the craniotomy consisted of drilling the mid-parietal skull bone to insert an extradural bolt, through which the brain injury was administered via fluid percussion from a pendulum device. An additional craniotomy was performed for the insertion of an intracranial pressure probe (ICP). The animals received TBI at time 0 (T0). SHAM animals were instrumented but remained uninjured.

### 4.3. Experimental Protocol

Animals were randomly allocated to receive either TBI or no injury (SHAM) and either 1 mL/kg IV NVX-428 (NVX) or normal saline (NON) such that there were four groups: No TBI and no NVX-428 (SHAM-NON; N = 4); No TBI and 1 mL/kg IV NVX-428 (SHAM-NVX; N = 4); TBI and no NVX-428 (TBI-NON; N = 6); TBI and 1 mL/kg IV NVX-428 (TBI-NVX; N = 8). Animals not receiving NVX received the same volume of 0.9% normal saline (NS) instead.

Baseline (T0) measurements were followed by TBI (or no injury), followed by a 15-min delay (T15) prior to NVX treatment (or no treatment with normal saline) to simulate arrival of medics. T75 simulated hospital arrival, where 1 g/kg IV mannitol could be administered (limited to two doses) if ICP > 20 mm Hg and mean arterial pressure (MAP) > 40 mm Hg. Hemodynamic (heart rate [HR], MAP, mean pulmonary artery pressure [MPAP], central venous pressure [CVP], cardiac index [CI] and pulmonary and systemic vascular resistance indexes [PVRI and SVRI]), and neurological (ICP, brain oxygenation [PbtO_2_] and cerebral perfusion pressure [CPP]) measurements were recorded. Arterial and venous blood samples were collected at regular intervals.

The animals were euthanized by IV injection of Beuthansia (10 mg/kg). Following euthanasia at T360, the brain was fixed in 10% buffered formalin for ≥48 h prior to paraffin embedding. Brain sections (cranial (rostral) cerebrum parietal lobe, cranial (rostral) and caudal hippocampus, and cranial (rostral) brainstem/cerebellum) were stained with H&E or Fluoro-Jade-B to identify degenerating neurons [29,30].

Slides were analyzed by a board-certified veterinary pathologist blinded to treatments. Hematoxylin and Eosin (H&E) sections were scored from 1 to 5 for mild to severe for hemorrhage, spongiosis, rarefaction, red/dead neurons, edema, inflammation, and cavitation based on a global visual microscopic assessment. These scores were then summed into an overall injury score. Regional assessments focused on spongiosis and ischemic (red)/dead neurons in entorhinal cortex plus subiculum (collectively referred to as “cerebrum” with regard to regional assessments), hippocampus, and cerebellum. For the regional hippocampal assessment, the upper (dorsal) hippocampus, not the ventrolateral portion, was assessed. Regional assessments of the entorhinal cortex + subiculum (cerebrum) were performed on both the upper and lower (dorso-mesial and ventro-lateral respectively) segments of those structures.

### 4.4. Statistical Analysis

Brain histopathology was the primary endpoint. Secondary endpoints included neurophysiological (i.e., ICP, CPP) and hemodynamic (i.e., CI, MPAP) measurements.

Similar to previous studies [10,28], an omnibus 2 (Injury) × 2 (Group) × 8 (time points) mixed-model ANOVA compared the physiological and blood gas data using IBM SPSS Statistics 23.0 (IBM Corporation, Armonk, NY). *t*-Tests determined which time points were different between groups. Then, a one-way repeated measure ANOVA of TBI-NVX and SHAM-NVX groups determined whether time affected measurements. MPAP, temperature and injury force were analyzed using ANOVA followed by post hoc Least Significant Difference to evaluate effect of drug treatment over time. For hematology, independent and paired *t*-tests compared groups. For histopathology, mean total injury score and individual component scores were analyzed using ANOVA to determine specific differences between groups. Cerebellar Fluoro-Jade B positive cell counts were analyzed using Student-*t*-test.

## 5. Conclusions

A 1.0 mL/kg IV dose of NVX-428 administered 15 min after TBI resulted in decreased damage to cerebellar Purkinje cells, neurons that are susceptible to hypoxemia. Furthermore, NVX-428 did not adversely affect blood pressures, cardiac index, intracranial pressure, or cerebral perfusion pressure in these brain-injured animals. The rapid but transient increase in mean pulmonary artery pressure after NVX-428 was not deemed to be detrimental in this TBI model. Additional investigations are needed to determine if NVX-428, a low-volume oxygen therapeutic, can provide protection against TBI by increasing brain tissue oxygenation for treating polytrauma patients with TBI and hemorrhagic shock in the pre-hospital environment. This study is relevant to the military as it could help reduce resuscitation fluid and increase oxygen delivery.

## Figures and Tables

**Figure 1 medsci-08-00041-f001:**
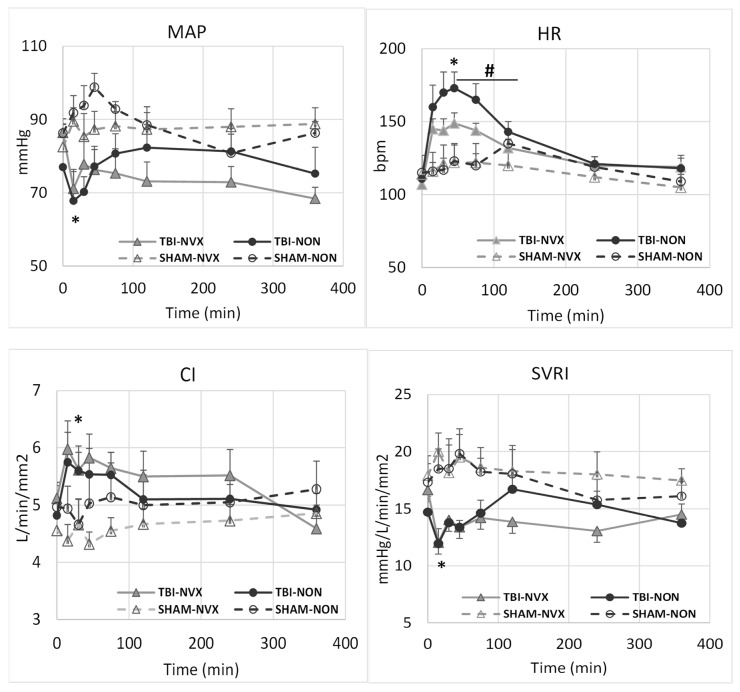
Mean arterial pressure (MAP), heart rate (HR), cardiac index (CI) and systemic vascular resistance index (SVRI) for the TBI-NVX, TBI-NON, SHAM-NVX and SHAM-NON groups. * *p* <0.01; significant difference between TBI vs. SHAM (combined TBI-NVX and TBI-NON vs. SHAM-NVX and SHAM-NON) indicating the initial physiological effect due to trauma. # *p* < 0.05; significant difference between TBI-NVX vs. TBI-NON or SHAM-NVX vs. SHAM-NON indicating the effect of NVX. Mean and standard deviation; SHAM-NON (--○--), SHAM-NVX (--∆--), TBI-NON (─●─), TBI-NVX (─▲─).

**Figure 2 medsci-08-00041-f002:**
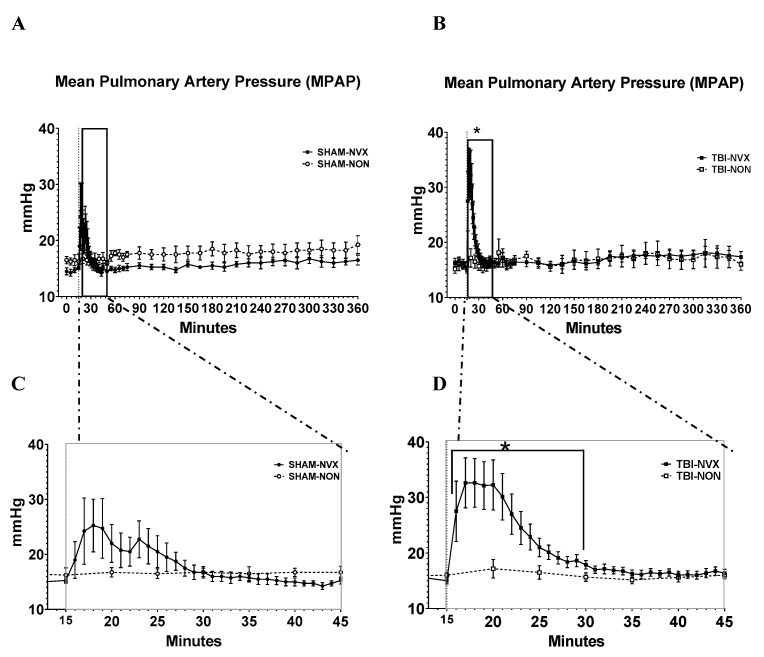
Mean pulmonary arterial pressure. Data (mean ± SEM) for the entire observation period (**A**,**B** panels, SHAM and TBI groups respectively) and expanded data for the first 45 min (**C**,**D** panels, SHAM and TBI groups respectivelyTBI-NVX was different from TBI-NON) between T15 and T30 (* *p <* 0.05); there were no differences between the SHAM groups.

**Figure 3 medsci-08-00041-f003:**
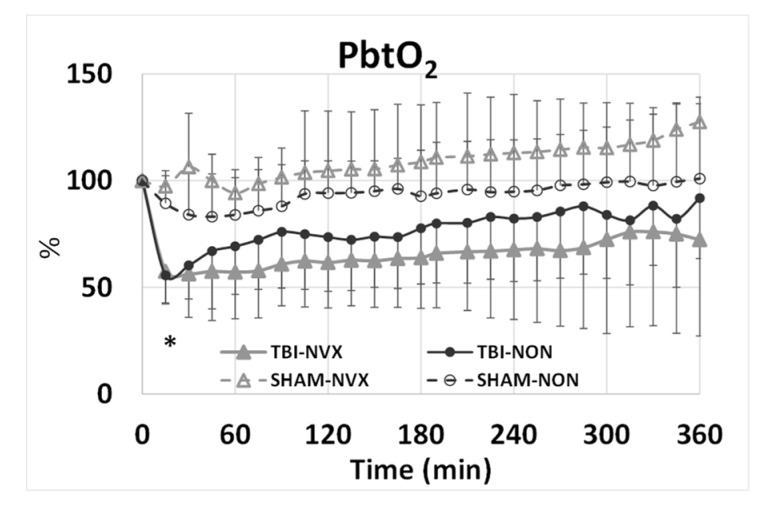
Brain tissue oxygenation (PtbO_2_) for the TBI-NVX, TBI-NON, SHAM-NVX and SHAM-NON groups. * *p* < 0.01; significant difference between TBI vs. SHAM (effect due to trauma). Mean and standard deviation; SHAM-NON (--○--), SHAM-NVX (--∆--), TBI-NON (─●─), TBI-NVX (─▲─).

**Figure 4 medsci-08-00041-f004:**
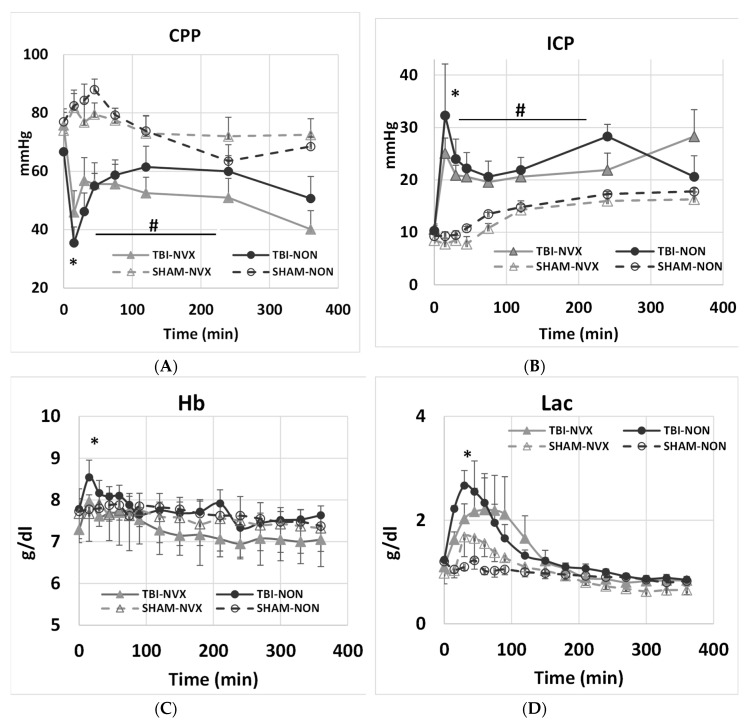
(**A**) Intracranial pressure (ICP) and (**B**) cerebral perfusion pressure (CPP) for the TBI-NVX, TBI-NON, SHAM-NVX and SHAM-NON groups. (**C**) Hb and (**D**) Lactate for the TBI-NVX, TBI-NON, SHAM-NVX and SHAM-NON groups. * *p* < 0.01; significant difference at T15 between TBI vs. SHAM (combined TBI-NVX and TBI-NON vs. SHAM-NVX and SHAM-NON) indicating the initial physiological effect due to TBI. Mean and standard deviation; SHAM-NON (--○--), SHAM-NVX (--∆--), TBI-NON (─●─), TBI-NVX (─▲─). # *p* < 0.05; significant difference from T15 to T200 for TBI groups.

**Figure 5 medsci-08-00041-f005:**
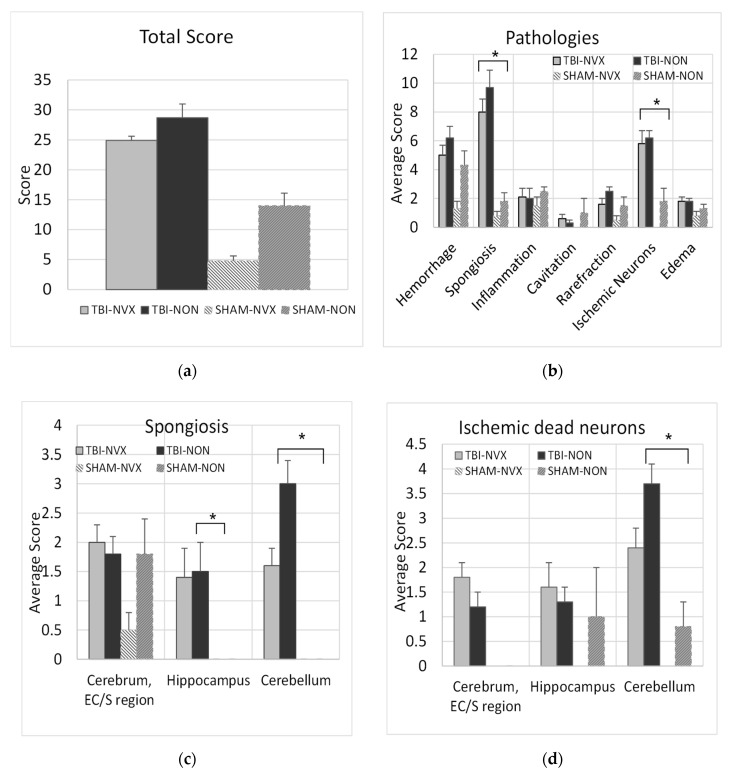
Pathology scores for various sections of the brain. (**a**) Global injury score from each pathology detected in brain sections (mean ± SEM) for all brain regions in swine with or without traumatic brain injury (TBI, SHAM, respectively) that received NVX-428 (NVX) or normal saline (NON) at 15 min after baseline and euthanized 360 min after baseline. (**b**) Break down from the various brain pathologies observed in different groups. (**c**) Specific regional injury scores for spongiosis and (**d**) ischemic/dead neurons (mean ± SEM) using Hematoxylin and Eosin (H&E) staining. EC/S = entorhinal cortex/subiculum. * *p* < 0.01 significance between SHAM and TBI groups.

**Figure 6 medsci-08-00041-f006:**
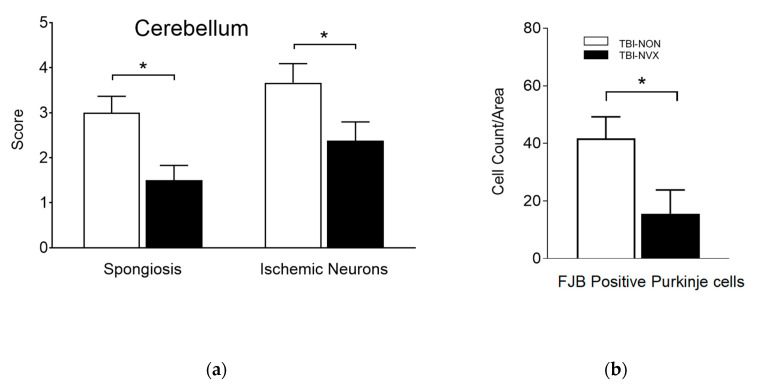
Regional area and severity score for cells stained with Fluoro-Jade B: cerebellar injury scores and cell counts. (**a**) Injury scores for spongiosis and ischemic/dead neurons and (**b**) cell counts (right panel, for Fluoro-Jade B positive Purkinje cells. Mean ± SEM; * *p* < 0.05 significant difference between treatment groups.

**Figure 7 medsci-08-00041-f007:**
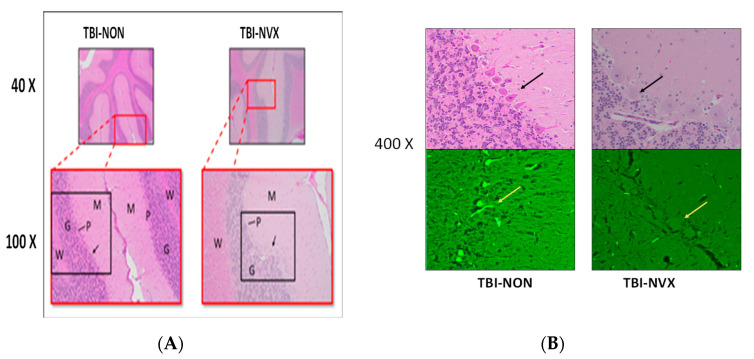
(**A**) Histopathology of cerebellar brain tissue. H&E stain from 40× to 400×. Red rectangles delineate the 40× areas that are further magnified by 100×. Black rectangles delineate the areas which are magnified by 200×. G = granular cell layer, M = molecular cell layer, *p* = Purkinje cell layer, and W = white matter; shrunken, pyknotic, and eosinophilic staining of the Purkinje cells from the TBI-NON animal. Black arrows point to the Purkinje cells further magnified at 400× in (**B**); (**B**) High magnification (400×) of framed panel in (**A**); cerebellar brain tissue using H&E (top) and Fluoro-Jade B (FJB, bottom). The FJB stains at 400× magnification are from adjacent sections to the respective H&E stains above them. Yellow arrow (FJB) points to adjacent Purkinje cells; note the fluorescent staining of the Purkinje cells in the TBI-NON animal and not in the TBI-NVX.

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
