# Peer review of "Treatment of Swine Closed Head Injury with Perfluorocarbon NVX-428"

_medsci, 2020, doi:10.3390/medsci8040041_

Round 1

Reviewer 1 Report

This study examined the safety and efficacy of the oxygen-carrier compound, NVX-428, in a swine TBI model.  The authors should commended for the work and effort required to undertake such as study..  The manuscript is generally well written, but I have highlighted some areas that need to be improved.  In addition, it appears the authors may have inadvertently mislabelled some of the figures.

Abstract

Line 20: Is it worth including :”NVX-428 (dodecafluoropentane emulsion; 2% w/v)”  in abstract so that readers unfamiliar with this compound are provided with this information from the start.

Line 23: replace “no-treatment” with  “normal saline”.   And throughout manuscript.

Line 24: replace “nothing” with “normal saline”.

Introduction

Lines 57-58: sentence “At body temperature, the droplets are postulated to expand slightly, providing enhanced oxygen transport ability [13].”   Is it worth mentioning/explaining why/how expansion of droplets enhances oxygen transport ability?  Is sentence necessary

Lines 70-72: sentence “These data and product characteristics suggest that NVX-428 may be suitable for the pre-hospital environment, particularly in areas where refrigeration is not immediately available, such as in military conflicts”   Storage of material during clinical trial is not the same as storage in field situation without refrigeration. Have any stability studies been performed where NVX-428 has been stored at room temperature for extended periods (e.g. 1-3 months) and the integrity of the product confirmed.

Results

Line 109: Why isn’t there a graph for the PbtO2 results? 

Line 112/113: CPP: provide name in full.  May be useful to provide all abbreviations in full in text, when first mentioned.

Line 145-151: Figure 3, figure legend.  Define  “*”.

Line 156: Sentence  “Total brain injury scores and subcategories of injury for the total brain were not different between these two groups (Fig 4)” Please specify in which figures in Fig 4; e.g. Fig 4 ab.  Note: Please specify other specific figures when appropriate in manuscript.

What’s the difference in Fig 4c (spongiosis) and d (ischemic/dead neurons) for cerebellum and Fig 5 spongiosis and ischemic/dead neurons for cerebellum.  I’m confused.

Line 159-165. Also,  in Fig 5  TBI NVX-428 treated animals showed greater spongiosis and ischemic/dead neurons and higher Fluoro-Jade B positive cells positive cells???  Is this correct??

Fig 6a.  Ideally,  magnification should be higher; why not provide at 100x and 400x, rather than 40x and 100x.

Fig 4.  On some occasions, Non-TBI animals appeared to display injury; this needs to be mentioned in results.  Good that it is mentioned/discussed in the Discussion (line 217 to 220).

Fig 5.  For consistency and to help the reader, provide label for Purkinje cell graph; “Purkinje cells”.

Discussion

Line 199:  sentence: “Furthermore, Fluoro-jade B staining of cerebrocortical neurons, hippocampal neurons, and cerebellar Purkinje cells was observed following TBI,..”  No Fluoro-jade B data is provided for cerebrocortical and, hippocampal neurons.

Line 216: “…and ischemic/dead neurons and decreased the number of Flouro-Jade B positive labeled cells.” – please specify which cells.

Line 220: Sentence “The results of this traumatic brain injury study indicate that NVX-428 may be a medically useful oxygen carrying compound under a variety of non traumatic conditions including stroke, myocardial infarction, anemia, and hypoxic solid tumors [5-7, 9, 11, 18, 26] .” Improve/justify  sentence; NVX-428 showing effects under traumatic conditions does not mean it is useful under “non traumatic” conditions.

MMs

The TBI model is not described???

It is strange to have the Conclusion section after the MMs section.  Shouldn’t it be at the end of the Discussion.

Minor

Line 375: “PbtO2: Tissue oxygenation” should this be – “Brain tissue oxygenation”

Author Response

REVIEWER #1

The authors want to thank the reviewer for his/her helpful comments.

See response in bold below each comments.

Abstract

Line 20: Is it worth including :”NVX-428 (dodecafluoropentane emulsion; 2% w/v)”  in abstract so that readers unfamiliar with this compound are provided with this information from the start.

This has been added.

Line 23: replace “no-treatment” with  “normal saline”.   And throughout manuscript.

Line 24: replace “nothing” with “normal saline”.

The text has been replaced.

Introduction

Lines 57-58: sentence “At body temperature, the droplets are postulated to expand slightly, providing enhanced oxygen transport ability [13].”   Is it worth mentioning/explaining why/how expansion of droplets enhances oxygen transport ability?  Is sentence necessary

We have added a short explanation through which this expansion enhances oxygen transport. We have also added an additional reference.

 Lines 70-72: sentence “These data and product characteristics suggest that NVX-428 may be suitable for the pre-hospital environment, particularly in areas where refrigeration is not immediately available, such as in military conflicts”   Storage of material during clinical trial is not the same as storage in field situation without refrigeration. Have any stability studies been performed where NVX-428 has been stored at room temperature for extended periods (e.g. 1-3 months) and the integrity of the product confirmed.

Stability studies have been performed by the company and support 9-months of room temperature storage and 2-years of refrigerated storage.

Results

Line 109: Why isn’t there a graph for the PbtO2 results? 

 We apologize for this omission. A graph has been added as Figure 3.

Line 112/113: CPP: provide name in full.  May be useful to provide all abbreviations in full in text, when first mentioned.

This was added in first mentioned and abbreviations.

Line 145-151: Figure 3, figure legend.  Define  “*”.

Significance level was updated in all graphs.

 Line 156: Sentence  “Total brain injury scores and subcategories of injury for the total brain were not different between these two groups (Fig 4)” Please specify in which figures in Fig 4; e.g. Fig 4 ab.  Note: Please specify other specific figures when appropriate in manuscript.

This text and the figure legends have been clarified.

What’s the difference in Fig 4c (spongiosis) and d (ischemic/dead neurons) for cerebellum and Fig 5 spongiosis and ischemic/dead neurons for cerebellum.  I’m confused.

Line 159-165. Also,  in Fig 5  TBI NVX-428 treated animals showed greater spongiosis and ischemic/dead neurons and higher Fluoro-Jade B positive cells positive cells???  Is this correct??

 Thank you for pointing out this discrepency. Fig 5 is a selection from Fiig 4c but  there was an error in the graph legend which was incorrect. It has now been changed.

Fig 6a.  Ideally,  magnification should be higher; why not provide at 100x and 400x, rather than 40x and 100x.

 The figures with higher magnification have been added and the legend has been updated.

Fig 4.  On some occasions, Non-TBI animals appeared to display injury; this needs to be mentioned in results.  Good that it is mentioned/discussed in the Discussion (line 217 to 220).

The severity score in SHAM-NON groups reflects that surgical  instrumentation necessary for performing TBI induces some damage. This is reported in the results and in the discussion text.

 Fig 5.  For consistency and to help the reader, provide label for Purkinje cell graph; “Purkinje cells”.

 This has been changed in the graph.

Discussion

Line 199:  sentence: “Furthermore, Fluoro-jade B staining of cerebrocortical neurons, hippocampal neurons, and cerebellar Purkinje cells was observed following TBI,..”  No Fluoro-jade B data is provided for cerebrocortical and, hippocampal neurons.

This sentence was corrected.

 Line 216: “…and ischemic/dead neurons and decreased the number of Flouro-Jade B positive labeled cells.” – please specify which cells.

This sentence was corrected to specify Purkinje cells.

Line 220: Sentence “The results of this traumatic brain injury study indicate that NVX-428 may be a medically useful oxygen carrying compound under a variety of non traumatic conditions including stroke, myocardial infarction, anemia, and hypoxic solid tumors [5-7, 9, 11, 18, 26] .” Improve/justify  sentence; NVX-428 showing effects under traumatic conditions does not mean it is useful under “non traumatic” conditions.

 This is an important point. Although we are lacking a true naïve control in this study, the Sham group provides a support for safety. The sentence has been modified.

MMs

The TBI model is not described???

We apologize for this omission. The TBI by fluid percussion is now described in methods (p 16).

It is strange to have the Conclusion section after the MMs section.  Shouldn’t it be at the end of the Discussion.

It is a format that is accepted by the journal

Minor

Line 375: “PbtO2: Tissue oxygenation” should this be – “Brain tissue oxygenation”

some of the figures.

The authors want to thank the reviewer for his/her helpful comments.

See response in bold below each comments.

Abstract

Line 20: Is it worth including :”NVX-428 (dodecafluoropentane emulsion; 2% w/v)”  in abstract so that readers unfamiliar with this compound are provided with this information from the start.

This has been added.

Line 23: replace “no-treatment” with  “normal saline”.   And throughout manuscript.

Line 24: replace “nothing” with “normal saline”.

The text has been replaced.

Introduction

Lines 57-58: sentence “At body temperature, the droplets are postulated to expand slightly, providing enhanced oxygen transport ability [13].”   Is it worth mentioning/explaining why/how expansion of droplets enhances oxygen transport ability?  Is sentence necessary

We have added a short explanation through which this expansion enhances oxygen transport. We have also added an additional reference.

 Lines 70-72: sentence “These data and product characteristics suggest that NVX-428 may be suitable for the pre-hospital environment, particularly in areas where refrigeration is not immediately available, such as in military conflicts”   Storage of material during clinical trial is not the same as storage in field situation without refrigeration. Have any stability studies been performed where NVX-428 has been stored at room temperature for extended periods (e.g. 1-3 months) and the integrity of the product confirmed.

Stability studies have been performed by the company and support 9-months of room temperature storage and 2-years of refrigerated storage.

Results

Line 109: Why isn’t there a graph for the PbtO2 results? 

 We apologize for this omission. A graph has been added as Figure 3.

Line 112/113: CPP: provide name in full.  May be useful to provide all abbreviations in full in text, when first mentioned.

This was added in first mentioned and abbreviations.

Line 145-151: Figure 3, figure legend.  Define  “*”.

Significance level was updated in all graphs.

 Line 156: Sentence  “Total brain injury scores and subcategories of injury for the total brain were not different between these two groups (Fig 4)” Please specify in which figures in Fig 4; e.g. Fig 4 ab.  Note: Please specify other specific figures when appropriate in manuscript.

This text and the figure legends have been clarified.

What’s the difference in Fig 4c (spongiosis) and d (ischemic/dead neurons) for cerebellum and Fig 5 spongiosis and ischemic/dead neurons for cerebellum.  I’m confused.

Line 159-165. Also,  in Fig 5  TBI NVX-428 treated animals showed greater spongiosis and ischemic/dead neurons and higher Fluoro-Jade B positive cells positive cells???  Is this correct??

 Thank you for pointing out this discrepency. Fig 5 is a selection from Fiig 4c but  there was an error in the graph legend which was incorrect. It has now been changed.

Fig 6a.  Ideally,  magnification should be higher; why not provide at 100x and 400x, rather than 40x and 100x.

 The figures with higher magnification have been added and the legend has been updated.

Fig 4.  On some occasions, Non-TBI animals appeared to display injury; this needs to be mentioned in results.  Good that it is mentioned/discussed in the Discussion (line 217 to 220).

The severity score in SHAM-NON groups reflects that surgical  instrumentation necessary for performing TBI induces some damage. This is reported in the results and in the discussion text.

 Fig 5.  For consistency and to help the reader, provide label for Purkinje cell graph; “Purkinje cells”.

 This has been changed in the graph.

Discussion

Line 199:  sentence: “Furthermore, Fluoro-jade B staining of cerebrocortical neurons, hippocampal neurons, and cerebellar Purkinje cells was observed following TBI,..”  No Fluoro-jade B data is provided for cerebrocortical and, hippocampal neurons.

This sentence was corrected.

 Line 216: “…and ischemic/dead neurons and decreased the number of Flouro-Jade B positive labeled cells.” – please specify which cells.

This sentence was corrected to specify Purkinje cells.

Line 220: Sentence “The results of this traumatic brain injury study indicate that NVX-428 may be a medically useful oxygen carrying compound under a variety of non traumatic conditions including stroke, myocardial infarction, anemia, and hypoxic solid tumors [5-7, 9, 11, 18, 26] .” Improve/justify  sentence; NVX-428 showing effects under traumatic conditions does not mean it is useful under “non traumatic” conditions.

 This is an important point. Although we are lacking a true naïve control in this study, the Sham group provides a support for safety. The sentence has been modified.

MMs

The TBI model is not described???

We apologize for this omission. The TBI by fluid percussion is now described in methods (p 16).

It is strange to have the Conclusion section after the MMs section.  Shouldn’t it be at the end of the Discussion.

It is a format that is accepted by the journal

Minor

Line 375: “PbtO2: Tissue oxygenation” should this be – “Brain tissue oxygenation”

This has been changed

Reviewer 2 Report

The manuscript describes an experiment evaluating the effects of an experimental oxygen carrier after a fluid percussion injury model of TBI in swine. The experimental model has high face validity for a clinical trauma population, and provides data supporting a protective effect of NVX-428 when given early after TBI. There are a few places where clarifications are needed to make the research easier to follow. Overall, the study is well done, with promising results.

Specific points:

- How many swine were in each group?

- Results, line 109-110: Where was PbO2 measured? In the discussion it is mentioned that there is lack of PbO2 data in this study as well. Was this measured over time? And if so, it might be interesting to see graphs of this, such as are shown for other physiologic variables as in Figs 1-3.

- Why were analyses done with both TBI groups combined in Fig 1 and Fig 3 data? Unless these have not been reported as changes in this particular swine model, I would suggest leaving this out (unless the authors mean there is a main effect of TBI in the analysis). Since the main point of the article appears to be whether there is a protective effect of NVX-428 on outcomes after TBI in this model, combining NVX and NON groups for this analysis would seem to undercut the main approach.

- It would be helpful to include more information from statistical analysis in the results section, such as F ratios (ie standard ANOVA F notation with degrees of freedom).

- Discussion, page 8 last paragraph, lines 225-227: discussing a difference between the half life of the drug and the physiologic effect. In the introduction, though, half-life was reported as 90 minutes, which is also how long the physiologic effects are being discussed as. I’m not sure why there is a reason to posit tissue deposition etc. to explain the effects, since after a single half-life it is reasonable to expect an ongoing effect of the treatment.

- Methods: needs a description of the FPI model.

- How was euthanasia performed?

Minor points:

- Introduction, line 54: what species was NVX-428 half-life determined in?

- In graphs in Fig 1 and Fig 3, it is difficult to distinguish NVX from NON groups. It might help to make the symbols different between groups (e.g. circle vs square), or make the gray more different from the black.

- In histopathology, page 5 line 135-136, TBI-NON is used twice; I believe the authors may have meant TBI vs sham, or TBI-NON vs sham-NON?

- Page 6, Fig 4 legend, b) I am not sure what “overall total injury score and overall subcategory injury score” means exactly, as there is only a single score for each group. Does this mean a summation of the total score and total of the subregions? Please clarify.

Author Response

REVIEWER #2

The authors want to thank the reviewer for his/her helpful comments.

See response in bold below each comments.

Specific points:

- How many swine were in each group?

This is in the methods (SHAM-NON; N = 4); (SHAM-NVX; N = 4); (TBI-NON; N = 6); (TBI-NVX; N = 8) (see page 16).

- Results, line 109-110: Where was PbO2 measured?

In methods: the Licox probe was inserted in the parietal side of the brain.

In the discussion it is mentioned that there is lack of PbO2 data in this study as well. Was this measured over time? And if so, it might be interesting to see graphs of this, such as are shown for other physiologic variables as in Figs 1-3.

 We apologize for this omission. A graph illustrating PbtO2 has been added.

- Why were analyses done with both TBI groups combined in Fig 1 and Fig 3 data? Unless these have not been reported as changes in this particular swine model, I would suggest leaving this out (unless the authors mean there is a main effect of TBI in the analysis). Since the main point of the article appears to be whether there is a protective effect of NVX-428 on outcomes after TBI in this model, combining NVX and NON groups for this analysis would seem to undercut the main approach.

- It would be helpful to include more information from statistical analysis in the results section, such as F ratios (ie standard ANOVA F notation with degrees of freedom).

The aim of performing this combination was to indicate the initial physiological effect due to TBI at the beginning of the experiment where no treatment was received. A sentence has been added. Statistical significance has been added.

- Discussion, page 8 last paragraph, lines 225-227: discussing a difference between the half life of the drug and the physiologic effect. In the introduction, though, half-life was reported as 90 minutes, which is also how long the physiologic effects are being discussed as. I’m not sure why there is a reason to posit tissue deposition etc. to explain the effects, since after a single half-life it is reasonable to expect an ongoing effect of the treatment.

We agree with this comment. The sentence has been removed.

- Methods: needs a description of the FPI model.

We apologize for this omission. The TBI by fluid percussion is now described in methods (p 16).

- How was euthanasia performed?

This was added to the methods; the animals were euthanized by IV injection of Beuthansia.

Minor points:

- Introduction, line 54: what species was NVX-428 half-life determined in?

This has been added

- In graphs in Fig 1 and Fig 3, it is difficult to distinguish NVX from NON groups. It might help to make the symbols different between groups (e.g. circle vs square), or make the gray more different from the black.

The graphs have been modified

- In histopathology, page 5 line 135-136, TBI-NON is used twice; I believe the authors may have meant TBI vs sham, or TBI-NON vs sham-NON?

Thank you. This error was corrected and the text was modified.

- Page 6, Fig 4 legend, b) I am not sure what “overall total injury score and overall subcategory injury score” means exactly, as there is only a single score for each group. Does this mean a summation of the total score and total of the subregions? Please clarify.

This was clarified in the legend and the text. The sequence of the figures was also altered.